# Does REM Sleep-Dependent Obstructive Sleep Apnea Have Clinical Significance?

**DOI:** 10.3390/ijerph192114147

**Published:** 2022-10-29

**Authors:** Seung Cheol Lee, Doh-Eui Kim, Young Hwangbo, Mei Ling Song, Kwang Ik Yang, Yong Won Cho

**Affiliations:** 1Sleep Disorders Center, Department of Neurology, Soonchunhyang University College of Medicine, Cheonan Hospital, 31, Suncheonhyang 6-gil, Dongnam-gu, Cheonan-si 31151, Korea; 2Department of Preventive Medicine, Soonchunhyang University College of Medicine, Cheonan Hospital, 31, Suncheonhyang 6-gil, Dongnam-gu, Cheonan-si 31151, Korea; 3Department of Nursing Science, College of Nursing, Daegu Health College, 15, Yeongsong-ro, Buk-gu, Daegu 41453, Korea; 4Department of Neurology, School of Medicine, Keimyung University, 1095 Dalgubeol-daero, Dalseo-gu, Daegu 42601, Korea

**Keywords:** REM sleep, obstructive sleep apnea, polysomnography, Korea

## Abstract

(1) Background: The clinical significance of rapid eye movement (REM) sleep-dependent obstructive sleep apnea (OSA) remains controversial because various criteria have been used to describe it. This study determined the clinical significance of REM-OSA in Koreans using data from patients with sufficient total sleep time (TST) and REM sleep duration. (2) Methods: We investigated 1824 patients with OSA who were diagnosed by polysomnography (PSG). REM-OSA was defined as an overall apnea–hypopnea index (AHI) ≥ 5, NREM-AHI < 15, and REM-AHI/NREM-AHI ≥ 2. Demographic and medical data were collected from digital medical records and sleep questionnaires. We compared clinical and PSG data between REM-OSA and REM sleep-nondependent OSA (nREM-OSA). (3) Results: In total, 140 patients (20.2%) were categorized as REM-OSA. Those patients were predominantly female (53.6% vs. 21.7% of the overall cohort, *p* < 0.001). REM-OSA is frequent in the mild (69.3% vs. 18.8%) to moderate (30% vs. 27.9%) range of OSA (*p* < 0.001). (4) Conclusions: The prevalence of REM-OSA was similar to that in previous study findings: frequent in mild to moderate OSA and females, which is consistent with results in Western populations. Our findings suggest that REM-OSA does not have clinical significance as a subtype of OSA.

## 1. Introduction

Rapid eye movement (REM) sleep-dependent obstructive sleep apnea (OSA) is diagnosed when obstructive apnea and hypopnea occur predominantly during REM sleep. A commonly used definition includes an overall apnea–hypopnea index (AHI) ≥ 5, with AHI in REM sleep (REM-AHI) that is twice that in non-REM (NREM) sleep (NREM-AHI) [1,2,3]. In Western countries, the prevalence of REM-OSA in clinical samples has been reported in a wide range, from 10% to 36% [1,2,3,4,5], because various criteria have been used to define it. However, previous studies have consistently shown that REM-OSA occurs more commonly in patients with mild or moderate OSA, females, young adults, and children [1,2,3,4,5]. A previous study about the prevalence of REM-OSA among Koreans showed that it accounted for 18% of OSA cases [6]. The effects of selective REM sleep fragmentation are unknown, and the clinical significance of REM-OSA remains controversial [7]. In a large community-based sample of middle-aged adults and seniors, REM-AHI was not independently associated with daytime sleepiness or quality of life after adjusting for NREM-AHI [8]. That finding was confirmed in a large clinical cohort [9]. However, those previous studies did not consider short TST or REM sleep duration. REM-AHI can be overestimated in individuals with short REM sleep duration and can introduce errors into the REM-AHI/NREM-AHI ratio. 

Our purposes in this study were to evaluate the prevalence of REM-OSA in a relatively large sample reflecting the clinical Korean population using data from patients with sufficient TST and REM sleep duration, to compare the clinical and polysomnographic data of patients with REM-OSA and REM sleep-nondependent OSA (nREM-OSA), and to identify factors that contribute to the observed differences.

## 2. Materials and Methods

We evaluated 1824 patients (≥18 years old) who visited sleep clinics and underwent polysomnography (PSG) in either of two hospitals in two different cities from 2012 to 2016. All subjects were Korean. Subjects were excluded for incomplete clinical data (*n* = 317), AHI < 5 (*n* = 524), total sleep time (TST) < 120 min (*n* = 16), total time spent in REM < 30 min (*n* = 225), and another sleep disorders in addition to OSA (28 restless legs syndrome, 7 narcolepsy, and 15 parasomnia). The remaining 692 patients were included in this study. This study received ethics approval from the institutional review board of each center. 

The following demographic and medical data were collected from digital medical records and sleep questionnaires: age, sex, body mass index (BMI), and results of the Korean versions of the Epworth Sleepiness Scale (K-ESS) [10], Pittsburgh Sleep Quality Index (K-PSQI) [11], Insomnia Severity Scale (K-ISI) [12], and Beck Depression Inventory-2 (K-BDI2) [13]. We recorded whether patients had a history of or current treatment for comorbid conditions of heart disease, diabetes, and hyperlipidemia. The heart diseases of the participants included coronary heart disease, valvular heart disease, arrhythmia, and congestive heart disease. 

Each patient underwent overnight PSG using a digital PSG (COMET PSG; Grass Technologies, Twin 4.5.2 Software, Warwick, RI, USA). The following variables were monitored: electroencephalogram (C3-A2, C4-A1, O2-A1, O1-A2), right and left electrooculogram, submental and both anterior tibialis electromyograms, electrocardiogram, airflow (pressure cannula and thermistor), respiratory effort (piezo-electric bands), oxyhemoglobin saturation (SaO_2_), and snoring. We used the 2012 American Academy of Sleep Medicine manual version 2.0 for respiratory event scoring [14]. The manual defines apnea as a ≥10-s decrease in the peak signal excursion ≥ 90% of the pre-event baseline, as recorded on an oronasal thermal sensor. Hypopnea was defined as a nasal pressure signal decrease of ≥30% of baseline that lasts for ≥10 s and is associated with either ≥3% oxygen desaturation or arousal. OSA severity was classified as mild (5 ≤ AHI < 15), moderate (15 ≤ AHI < 30), or severe (30 ≤ AHI). 

Patients were considered to have REM-OSA according to the definition in a previous report: overall AHI ≥ 5, NREM-AHI < 15, and REM-AHI/NREM-AHI ≥ 2 [3,6]. PSG data included TST, sleep latency, REM latency, sleep efficiency, wake after sleep onset (WASO), percentage of sleep stage (N1, N2, N3, and REM sleep), percentage of sleep duration in the supine position, arousal index, AHI, hypopnea index (HI), apnea index (AI), REM-AHI, NREM-AHI, supine AHI, off-supine AHI, mean O_2_ saturation (SaO_2_), and periodic limb movement (PLM) index. 

Data are expressed as the mean ± standard deviation for continuous variables and as numbers and percentages for categorical variables. Data were analyzed using the independent sample t-test for continuous variables and the χ^2^ test for categorical variables. Differences were considered statistically significant at a *p* value < 0.05. Data were analyzed using statistical software (SPSS for Windows 17.0, Chicago, IL, USA).

## 3. Results

The demographic and clinical characteristics of the entire group of OSA patients and the REM-OSA and nREM-OSA groups are shown in Table 1. Among the 692 OSA patients, 140 (20.2%) met the definition of REM-OSA, and 552 (79.8%) had nREM-OSA. In addition, 53.6% of the REM-OSA patients were female, whereas the nREM-OSA group had a lower prevalence of females (21.7%, *p* < 0.001). 

Compared with the REM-OSA group, patients with nREM-OSA had a higher prevalence of hypertension (*p* < 0.001) and diabetes (*p* = 0.016). There were no significant differences in BMI or prevalence of heart disease or hyperlipidemia between the REM-OSA and nREM-OSA groups. No differences were observed in the K-ESS, K-PSQI, K-ISI, or K-BDI2 scores between the groups. In the REM-OSA group, the number of patients decreased as OSA severity (*p* < 0.001) increased, with 97 (69.3%) patients having mild OSA, 42 (30.0%) having moderate OSA, and 1 (0.7%) having severe OSA. In the nREM-OSA group, in contrast, the number of patients increased as OSA severity increased, with 104 (18.8%) having mild OSA, 154 (27.9%) having moderate OSA, and 294 (53.2%) having severe OSA.

Table 2 shows the polysomnographic findings in the REM-OSA and nREM-OSA groups. Several parameters differed significantly between the groups. Compared with the nREM-OSA group, REM-OSA patients spent less time in REM latency (100.1 ± 62.0 vs. 110.5 ± 60.3), less sleep time in N1 sleep (16.4 ± 7.8 vs. 28.3 ± 13.2, *p* < 0.001), and less WASO (50.9 ± 39.7 vs. 65.5 ± 46.4, *p* < 0.001) and had a lower total arousal index (25.0 ± 10.8 vs. 41.8 ± 19.5, *p* < 0.001). The AHI, HI, AI, NREM AHI and supine AHI were all significantly lower in the REM-OSA group than the nREM-OSA group (12.4 ± 5.5 vs. 36.5 ± 22.6, *p* < 0.001; 11.0 ± 5.0 vs. 26.0 ± 15.7, *p* < 0.001; 1.4 ± 2.2 vs. 10.6 ± 16.2, *p* < 0.001; 7.6 ± 3.8 vs. 36.8 ± 23.7, respectively). 

Table 3 shows the clinical characteristics and PSG data of the REM-OSA and nREM-OSA groups according to OSA severity. As the REM-OSA group contained only one case of severe OSA, we did not analyze that category. In the group of mild OSA patients, the REM-OSA patients had a higher prevalence of females (58.8% vs. 31.7%, *p* < 0.001). The mild REM-OSA group had less WASO (51.3 ± 41.3 vs. 68.3 ± 50.2, *p* = 0.049), less time in N1 sleep (16.0 ± 7.5 vs. 18.9 ± 8.1, *p* = 0.042), less time in N3 sleep (4.3 ± 7.6 vs. 6.5 ± 9.0), less NREM AHI (5.9 ± 2.7 vs. 10.5 ± 3.4, *p* < 0.001), and less supine AHI (13.5 ± 8.7 vs. 21.6 ± 21.9, *p* < 0.001) than the mild nREM-OSA group. In the group of moderate OSA patients, the prevalence of female patients did not differ significantly between the REM-OSA and nREM-OSA groups. The REM-OSA group had less WASO (48.7 ± 35.7 vs. 62.0 ± 47.2, *p* = 0.046), less time in N1 sleep (17.4 ± 8.2 vs. 23.7 ± 10.7, *p* < 0.001), less AHI (18.9 ± 3.2 vs. 22.5 ± 4.3, *p* < 0.001), less HI (16.3 ± 3.7 vs. 19.8 ± 5.2, *p* < 0.001), less NREM AHI (11.4 ± 3.0 vs. 21.7 ± 5.0, *p* < 0.001), and less supine AHI (24.2 ± 20.6 vs. 36.0 ± 16.5, *p* < 0.001) than the moderate nREM-OSA group.

## 4. Discussion

In a Korean population from two hospitals, the prevalence of REM-OSA with different degrees of OSA severity was 20.2%. The prevalence of REM-OSA in previous Western studies ranged from 10% to 36% [1,2,3,4,5] due to heterogeneity in definition. The REM-OSA definition used here is overall AHI ≥ 5, NREM-AHI < 15, and REM AHI/NREM-AHI ≥ 2. This definition is problematic because it can include patients who experience OSA in NREM and REM sleep severe enough that REM-AHI/NREM-AHI ≥ 2. Moreover, this measure is imprecise because the ratio can be high because of either a high REM-AHI, a low NREM-AHI, or a combination of the two [7]. A previous large Western clinical population study compared the prevalence and clinical characteristics of REM-OSA defined with various criteria [5]. Although a stricter definition (AHI ≥ 5, REM AHI/NREM-AHI ≥ 2, NREM-AHI < 8, and at least 10.5 min of REM sleep duration) decreased the prevalence by approximately twofold compared with a definition of AHI ≥ 5, REM-AHI/NREM-AHI ≥ 2, and NREM-AHI < 15, the stricter definition did not appear to be more clinically useful in classifying patients. Therefore, we used the previously published definition of AHI ≥ 5, REM-AHI/NREM-AHI ≥ 2, and NREM-AHI < 15 in this study [2,3,5,6,15]. In agreement with our results, a previous Korean study found that REM-OSA accounted for 18% of OSA patients [6]. Although that population was from one hospital, the definition and sample size of that study were similar to ours. Estimates of the REM-AHI/NREM-AHI ratio can be highly imprecise if total sleep and REM sleep times are not considered. Previous studies [2,3] enrolled subjects who met additional criteria (TST > 100 min, REM sleep time > 10 min). In consideration of the threshold for REM sleep duration, estimates of REM-AHI can be imprecise if the duration of REM sleep is less than 30 min [7]. Thus, in selecting cases for this study, we added exclusion criteria (TST < 120 min, total time spent in REM < 30 min). 

We found that REM-OSA is more commonly diagnosed in patients with mild to moderate OSA than nREM-OSA, which is similar to the results of Western studies [2,3,5,15]. Our results are also similar to a previous Korean study [6]. Among REM-OSA patients, only 0.7% had severe OSA, whereas 53.2% of nREM-OSA patients had severe OSA. Most respiratory parameters, including AHI, AI, HI, and supine AHI were significantly worse in nREM-OSA patients than in the REM-OSA group, which is consistent with previous studies [1,16]. The average AHI in the REM-OSA group was one-third of that in the nREM-OSA group. The NREM-AHI in nREM-OSA patients was significantly higher than that in REM-OSA patients, but REM-AHI was similar in nREM-OSA and REM-OSA patients. The difference between the groups in AI was as high as eightfold. The supine AHI and off-supine AHI in nREM-OSA patients were significantly higher than those in REM-OSA patients, but the clinical significance of this finding is difficult to explain. In comparing the sleep architecture of the two groups, we found discrepancies among the results of other studies. One study found a significant difference in the duration of all stages except stage N2 sleep between the two groups [1]. Another study found no significant difference in the duration of REM sleep but more slow-wave sleep in the REM-OSA group [16]. In this study, most components of sleep architecture (except stage N1 sleep) showed no significant differences between the groups. Those discrepancies likely reflect our exclusion criteria (total time spent in REM < 30 min), which ensured that our patients had a longer duration of REM sleep than those in other studies. 

Despite the differences in exclusion criteria (TST, duration of REM sleep), we found that REM-OSA is more common in females, as in prior studies that used the same definition of REM-OSA [2,3,15]. Although only 28.2% of OSA patients overall were female in this study, 53.6% of the REM-OSA patients were female, compared with only 21.7% of the nREM-OSA patients. During NREM sleep, airway resistance increases less in females than in males [17]. Female hormones increase the tonicity of the genioglossus [18], and progesterone stimulates ventilation in women, men, and rats [19,20,21] and significantly increases the ventilator response to both hypoxia and hypercapnia [21]. As atonia occurs in REM sleep, female hormones might lose their ability to act on muscles in the upper airway, predisposing females to airway collapse because their upper airway dimensions tend to be smaller than those in males [22]. 

The age of REM-OSA and nREM-OSA patients did not differ significantly, which is different from previous findings [1,2,3,16]. Our definition of REM-OSA was the same as in those studies, but the sample selection process was different. We included only patients who had ≥120 min of TST and ≥30 min of REM sleep. Previous studies had no criteria for TST or duration of REM sleep [1] or had shorter limitations (TST ≥ 100 min or duration of REM sleep ≥ 10 min) [2,3,16]. 

The average BMI of the Korean population is lower than that of the American population [23,24,25]. Unlike previous studies [1,3], BMI did not differ between the REM-OSA and nREM-OSA groups in this study. It is uncertain whether the lower BMI caused an influence on this discrepancy. Compared with REM-OSA patients, more nREM-OSA patients had hypertension and diabetes, probably because the nREM-OSA group had more severe OSA and worse respiratory-related parameters than the REM-OSA group. Daytime sleepiness, sleep quality, insomnia symptoms, and depression did not differ significantly between the groups in this study even though REM and NREM sleep are functionally distinct states [26]. This finding is similar to the result of another smaller study [27]. We did not find any differences in the questionnaire results from either group according to OSA severity. Several cross-sectional clinic-based and prospective epidemiologic studies have suggested that only obstructive events during NREM sleep are associated with excessive daytime sleepiness or impairment of daily life [8,9,28]. REM sleep usually accounts for less than 30% of TST, with NREM sleep accounting for more than 70% of sleep time. Thus, the severity of OSA measured by NREM-AHI corresponds to a significantly higher number of apneas and hypopneas than REM-AHI. Moreover, as assessed by the mean sleep latency test, normal individuals selectively deprived of REM sleep manifested smaller changes in the degree of daytime sleepiness than subjects deprived of NREM sleep [29]. Thus, REM sleep disruption appears to have little effect on daytime sleepiness.

This study has some limitations. First, it is a cross-sectional retrospective study, so we cannot explain whether REM-OSA in a mild or moderate form is a natural course of OSA. Second, there is a possibility of night-to-night variability in AHI because all patients received PSG in a single night of sleep. Third, we did not evaluate the amount of time spent in each position during REM and NREM sleep. The supine position has a deleterious effect on sleep-related breathing disorders [30,31]. In this study, we found that the sleep time spent in the supine position did not differ significantly between the groups. Therefore, it is difficult to suggest that a difference in sleep position had an effect on respiratory events during REM and NREM sleep. Fourth, we did not investigate whether the patients were treated with CPAP or their clinical outcomes. Therefore, it would be good to study treatment plans with CPAP and clinical outcomes. Additionally, researching a critical definition that can make the clinical significance of REM-OSA can be better than conducting studies with the definition that previous studies suggested. Our clinical characteristics were based on electronic medical records and sleep questionnaires. Despite those limitations, our study has a large sample from two hospitals. Although our sample selection method differed from that in previous studies (the exclusion criteria in this study included TST < 120 min, total time spent in REM < 30 min.), our findings, except for age, are consistent with those of previous studies that used the same definition of REM-OSA.

## 5. Conclusions

In conclusion, although the selection criteria for this study required a longer duration of TST and REM sleep than those of previous studies, the prevalence of REM-OSA is similar to the finding of a previous Korean study. As shown before in Western studies, we also found that the REM-OSA group had more prevalence of females. Additionally, REM-OSA is more commonly diagnosed in patients with mild to moderate OSA than nREM-OSA in the Korean population. Daytime sleepiness, sleep quality, insomnia, and depressive mood did not differ significantly between the REM-OSA and nREM-OSA groups. Therefore, although REM-OSA shows a sex difference from nREM-OSA, our replicated findings suggest that REM-OSA does not have clinical significance as a subtype of OSA. However, because the definition of REM-OSA is ambiguous, we suggest the need for further research using various criteria for defining REM-OSA and comparing the differences in the results according to the duration of TST or REM sleep to estimate the clinical significance of REM-OSA.

## Figures and Tables

**Table 1 ijerph-19-14147-t001:** The demographic and clinical characteristics of the entire group and REM-OSA and nREM-OSA patients.

	Total (*n* = 692)	REM-OSA (*n* = 140)	nREM-OSA (*n* = 552)	*p* Value
Patients, %	100	20.2	79.8	
Female (%)	195 (28.2)	75 (53.6)	120 (21.7)	<0.001
Age, year	50.3 ± 13.4	48.7 ± 12.9	50.7 ± 13.4	NS
BMI, kg/m^2^	25.6 ± 3.8	25.4 ± 4.1	25.6 ± 3.8	NS
Heart disease (%)	98 (14.2)	15 (10.7)	83 (15.0)	NS
Hypertension (%)	227 (32.8)	29 (20.7)	198 (35.9)	0.001
Diabetes (%)	92 (13.3)	11 (7.1)	82 (14.9)	0.016
Hyperlipidemia (%)	104 (15.0)	17 (12.1)	87 (15.8)	NS
OSA severity				<0.001
5 ≤ AHI < 15	201 (29.0)	97 (69.3)	104 (18.8)	
15 ≤ AHI < 30	196 (28.3)	42 (30.0)	154 (27.9)	
AHI ≥ 30	295 (42.6)	1 (0.7)	294 (53.2)	
K-ESS	8.8 ± 4.7	8.7 ± 4.8	8.9 ± 4.7	NS
K-PSQI	8.1 ± 4.0	8.4 ± 3.9	8.0 ± 4.0	NS
K-ISI	12.1 ± 6.6	12.3 ± 6.9	12.0 ± 6.5	NS
K-BDI2	13.2 ± 9.4	14.7 ± 9.5	12.9 ± 9.3	NS

Data are presented as mean ± SD or number (%). REM-OSA, REM sleep-dependent obstructive sleep apnea; nREM-OSA, REM sleep-nondependent obstructive sleep apnea; BMI, body mass index; OSA, obstructive sleep apnea; AHI, apnea–hypopnea index; K-ESS, Korean version of the Epworth Sleepiness Scale; K-PSQI, Korean version of the Pittsburgh Sleep Quality Index; K-ISI, Korean version of the Insomnia Severity Scale; K-BDI2, Korean version of the Beck Depression Inventory 2.

**Table 2 ijerph-19-14147-t002:** Polysomnographic data for the entire group and REM-OSA and nREM-OSA patients.

	Total (*n* = 692)	REM-OSA (*n* = 140)	nREM-OSA (*n* = 552)	*p* Value
TST, min	35.1.3 ± 55.6	363.8 ± 49.0	348.1 ± 56.8	NS
Sleep latency, min	11.9 ± 21.0	14.7 ± 21.7	11.3 ± 20.7	NS
REM latency, min	108.6 ± 60.8	100.1 ± 62.0	110.8 ± 60.3	0.031
Sleep efficiency, %	82.6 ± 11.7	85.1 ± 10.5	82.0 ± 12.0	NS
WASO	62.5 ± 45.5	50.9 ± 39.7	65.5 ± 46.4	<0.001
N1, %	25.9 ± 13.3	16.4 ± 7.8	28.3 ± 13.2	<0.001
N2, %	52.5 ± 12.5	60.3 ± 9.3	50.5 ± 12.5	NS
N3, %	3.0 ± 6.8	3.7 ± 7.2	2.9 ± 6.7	NS
REM, %	18.6 ± 5.9	19.6 ± 5.9	18.2 ± 5.8	NS
Supine position, %	62.2 ± 27.7	68.0 ± 25.4	60.7 ± 28.1	NS
Total arousal index	38.4 ± 19.3	25.0 ± 10.8	41.8 ± 19.5	<0.001
AHI	31.6 ± 22.5	12.4 ± 5.5	36.5 ± 22.6	<0.001
HI	22.9 ± 15.4	11.0 ± 5.0	26.0 ± 15.7	<0.001
AI	8.7 ± 15.0	1.4 ± 2.2	10.6 ± 16.2	<0.001
REM AHI	34.2 ± 23.9	31.5 ± 15.9	35.9 ± 25.5	NS
NREM AHI	30.9 ± 24.2	7.6 ± 3.8	36.8 ± 23.7	<0.001
Supine AHI	43.2 ± 28.8	16.6 ±10.7	49.9 ±28.1	<0.001
Off-supine AHI	15.4 ± 21.3	5.3 ± 9.0	18.0 ± 22.6	<0.001
Mean SaO_2,_ %	94.8 ± 2.6	95.5 ± 2.9	94.6 ± 2.5	NS
PLM index	4.9 ± 12.6	5.6 ± 14.2	4.7 ± 12.2	NS

REM-OSA: REM sleep-dependent obstructive sleep apnea; nREM-OSA, REM sleep-nondependent obstructive sleep apnea; TST, total sleep time; WASO, wake after sleep onset; AHI, apnea-hypopnea index; HI, hypopnea index; AI, apnea index; SaO_2_, O_2_ saturation; PLM, periodic limb movements during sleep.

**Table 3 ijerph-19-14147-t003:** Clinical characteristics and polysomnographic data for REM-OSA and nREM-OSA patients according to OSA severity.

	Mild (*n* = 201)			Moderate (*n* = 196)		
Variables	REM-OSA (*n* = 97)	nREM-OSA (*n* = 104)	*p* Value	REM-OSA (*n* = 42)	nREM-OSA (*n* = 154)	*p* Value
Female (%)	57 (58.8)	33 (31.7)	<0.001	17 (40.5)	41 (26.6)	NS
Age, year	48.7 ± 12.7	48.7 ± 14.7	NS	48.5 ± 13.5	50.7 ± 12.8	NS
BMI, kg/m^2^	25.2 ± 4.0	24.2 ± 2.8	NS	26.1 ± 4.6	24.9 ± 3.4	NS
Heart disease (%)	11 (11.3)	13 (12.5)	NS	4 (9.5)	21 (13.6)	NS
Hypertension (%)	18 (18.6)	24 (23.1)	NS	10 (23.8)	54 (35.1)	NS
Diabetes (%)	7 (7.2)	11 (10.6)	NS	2 (4.8)	23 (15.0)	NS
Hyperlipidemia (%)	11 (11.3)	13 (12.5)	NS	6 (14.3)	22 (14.3)	NS
K-ESS	8.4 ± 5.1	8.2 ± 4.7	NS	9.2 ± 3.7	8.6 ± 4.8	NS
K-PSQI	8.4 ± 3.9	8.2 ± 4.0	NS	8.4 ± 3.7	8.1 ± 4.0	NS
K-ISI	12.5 ± 6.9	12.1 ± 6.4	NS	11.8 ± 7.0	12.0 ± 6.8	NS
K-BDI2	14.4 ± 9.0	14.0 ± 9.9	NS	15.2 ± 10.6	13.6 ± 9.2	NS
Polysomnography data						
TST, m	365.4 ±52.5	355.6 ± 57.5	NS	360.5 ± 41.1	354.6 ± 59.1	NS
Sleep latency, min	14.7 ± 21.0	12.2 ± 22.9	NS	14.5 ± 23.7	10.2 ± 12.2	NS
REM latency, min	106.8 ± 64.5	104.8 ± 57.6	NS	85.4 ± 54.2	108.7 ± 56.6	NS
Sleep efficiency	84.9 ± 10.9	81.8 ± 12.3	NS	85.9 ± 9.4	83.0 ± 11.5	NS
WASO	51.3 ± 41.3	68.3 ± 50.2	0.049	48.7 ± 35.7	62.0 ± 47.2	0.046
N1, %	16.0 ± 7.5	18.9 ± 8.1	0.042	17.4 ± 8.2	23.7 ± 10.7	<0.001
N2, %	60.6 ± 9.3	54.5 ± 11.9	NS	59.8 ± 9.2	54.0 ± 12.3	NS
N3, %	4.3 ± 7.6	6.5 ± 9.0	0.032	2.5 ± 6.3	3.5 ± 8.0	NS
REM, %	19.1 ± 5.8	20.1 ± 6.1	NS	20.2 ± 5.8	18.8 ± 5.8	NS
Supine position (%)	66.0 ± 25.5	53.6 ± 26.7	NS	74.3 ± 22.7	57.7 ± 29.3	NS
Arousal index	23.5 ± 10.7	24.4 ± 13.2	NS	28.1 ± 10.6	33.1 ± 12.5	0.009
AHI	9.4 ± 2.9	9.9 ± 3.0	NS	18.9 ± 3.2	22.5 ± 4.3	<0.001
HI	8.6 ± 2.8	9.0 ± 2.9	NS	16.3 ± 3.7	19.8 ± 5.2	<0.001
AI	0.9 ± 1.2	0.9 ± 1.2	NS	2.6 ± 3.3	2.8 ± 4.0	NS
REM AHI	24.7 ± 10.9	7.4 ± 5.5	NS	46.3 ± 14.4	27.0 ± 17.6	NS
NREM AHI	5.9 ± 2.7	10.5 ± 3.4	<0.001	11.4 ± 3.0	21.7 ± 5.0	<0.001
Supine AHI	13.5 ± 8.7	21.6 ± 21.9	<0.001	24.2 ± 20.6	36.0 ± 16.5	<0.001
Off-supine AHI	3.7 ± 4.7	3.7 ± 2.9	NS	8.5 ± 13.6	8.8 ± 8..2	NS
Mean SaO_2,_ %	95.5 ± 3.3	95.8 ± 1.7	NS	95.4 ± 1.4	95.3 ± 1.5	NS
PLM index	6.1 ± 15.6	6.6 ± 14.0	NS	4.5 ± 10.7	5.0 ± 13.0	NS

Data are presented as mean ± SD or number (%). REM-OSA, REM sleep-dependent obstructive sleep apnea; nREM-OSA, REM sleep-nondependent obstructive sleep apnea; BMI, body mass index; K-ESS, Korean version of the Epworth Sleepiness Scale; K-PSQI, Korean version of the Pittsburgh Sleep Quality Index; K-ISI, Korean version of the Insomnia Severity Scale; K-BDI2, Korean version of the Beck Depression Inventory 2; TST, total sleep time; WASO, wake after sleep onset; AHI, apnea-hypopnea index; HI, hypopnea index; AI, apnea index; SaO_2_, O_2_ saturation; PLM, periodic limb movements during sleep.

## Data Availability

The data that support the findings of this study are available on request from the corresponding author [K.I.Y.].

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
