# Peer review of "Does REM Sleep-Dependent Obstructive Sleep Apnea Have Clinical Significance?"

_ijerph, 2022, doi:10.3390/ijerph192114147_

Round 1
Reviewer 1 Report
Whether OSA that is obviously worse in REM sleep is a clinical entity remains debated. There are other cross sectional studies in this area similar to the authors study.
They have taken a sample of convenience with 1824 patients having in-patient psg seen across 2 big sleep services, then picking 692 of this group with OSA alone on PSG, no other sleep disorder and enough TST (>120 mins) and REM (>30 mins) to score. Then measured 20% as having REM-OSA.
They have set a reasonable but still somewhat arbitary definition of REM-OSA, not exactly the same as other papers. One confounding issue for many such cross sectional studies is that the definition itself will pre-determine many of the reported results, in particular the NREM-AHI being less than 15. Very severe OSA can cause far more movement and awakenings in REM making scoring difficult in this sleep stage. One potential reason that their group and others largely find this to be a less severe and less symptomatic group.
Key would be follow up of the REM-OSA group compared to NREM-OSA and reporting of some variables that seem likely to be within the datasets of 2 large sleep services, including medications that might affect REM, eg antidepressants, but also initial reason for referral, particularly as the mean ESS seems low and not that sleepy at 8, ie within normal range. Did the 2 tertiary clinics only perform in-patient psg or did they also use more limited respiratory sleep studies? And time frame that data was collected over is not stated.
Key would really be whether this group had progressed symptomatically or whether they were treated with CPAP and what their outcomes were, were they different from NREM-OSA?. This seems possible within study design and dataset to report and would really add to the paper and would add novelty to the data.
The results are similar to other large datasets with exception of lower BMI than would typically be reported in psg studies for OSA in a US or European population. BMI discrepancies not discussed. I do not know why they did not measure supine time - this is a standard outcome of all modern psg equipment and would be useful additional data, the authors acknowledge this as a limitation but don't give a reason.
Minor points include
1. Some contradictory statements in discussion in lines 210-212. They say groups same age and then different in subsequent sentences.
2. Very many grammatical errors relating to english use and I would suggest that the manuscript would benefit from proof reading by an editor with english as a first language, this is minor and the manuscript was comprehensible.
Author Response
Whether OSA that is obviously worse in REM sleep is a clinical entity remains debated. There are other cross sectional studies in this area similar to the authors study.
They have taken a sample of convenience with 1824 patients having in-patient psg seen across 2 big sleep services, then picking 692 of this group with OSA alone on PSG, no other sleep disorder and enough TST (>120 mins) and REM (>30 mins) to score. Then measured 20% as having REM-OSA.
They have set a reasonable but still somewhat arbitary definition of REM-OSA, not exactly the same as other papers. One confounding issue for many such cross sectional studies is that the definition itself will pre-determine many of the reported results, in particular the NREM-AHI being less than 15.
Response; In this study, we used same definition of REM-OSA as which previous studies had used. [1-3] But, duration of TST and REM sleep is longer than that of previous studies.
Researching into a critical definition, which makes clinical significance of REM-OSA, can be better than conducting studies with definition which previous studies suggested. Thank you for your good comment. We have revised the discussion of the manuscript and suggest it as future agenda for further research.
Very severe OSA can cause far more movement and awakenings in REM making scoring difficult in this sleep stage. One potential reason that their group and others largely find this to be a less severe and less symptomatic group.
Response; As you commented, scoring REM stage is difficult in case of very severe OSA. As a result, the duration of REM sleep stage can be short. Thank you for your good comment again. We entirely agree with you.
Key would be follow up of the REM-OSA group compared to NREM-OSA and reporting of some variables that seem likely to be within the datasets of 2 large sleep services, including medications that might affect REM, eg antidepressants, but also initial reason for referral, particularly as the mean ESS seems low and not that sleepy at 8, ie within normal range. Did the 2 tertiary clinics only perform in-patient psg or did they also use more limited respiratory sleep studies? And time frame that data was collected over is not stated.
Response; This is a retrospective study. In Korea, PSG has been provided to patients enrolled in the national health insurance since 2018. Before 2018, few primary health clinics could conduct PSG, so most occurred in tertiary clinics. Although our clinic is tertiary, medical systems differ by country in ways that render your comment moot. We seek your understanding for this difference and limitation of our dataset. We changed ‘tertiary university hospital’ to ‘hospital’ in the revised version to prevent misunderstanding.
The severity of OSA and severity of ESS do not correlate linearly, and in Korea, patients who present to a tertiary clinic for OSA treatment might not have severe clinical symptoms related to daytime sleepiness, and the average ESS score was below 10 or 11.
We evaluated patients who visited sleep clinics and underwent polysomnography in either of two hospitals in two different cities from 2012 to 2016.
Key would really be whether this group had progressed symptomatically or whether they were treated with CPAP and what their outcomes were, were they different from NREM-OSA? This seems possible within study design and dataset to report and would really add to the paper and would add novelty to the data.
Response; We did not investigate whether patients were treated with CPAP or their clinical outcomes. That would be a good future study if it is investigated and analyzed whether the patients were treated with CPAP and what their clinical outcomes were. In this study, unfortunately, we do not have the data you requested so we cannot provide it. However, we have revised the discussion of the manuscript and suggest further research into those results.
The results are similar to other large datasets with exception of lower BMI than would typically be reported in psg studies for OSA in a US or European population. BMI discrepancies not discussed. I do not know why they did not measure supine time - this is a standard outcome of all modern psg equipment and would be useful additional data, the authors acknowledge this as a limitation but don't give a reason.
Response; The average BMI in the Korean population is lower than in the American population. We have added that information to the discussion.
We present the proportion of sleep time in a supine position relative to the total sleep time. Unfortunately, sleep time in the supine position during REM sleep or NREM sleep is not included in our dataset.
Minor points include
- Some contradictory statements in discussion in lines 210-212. They say groups same age and then different in subsequent sentences.
Response; We revised the statements which you commented.
- Very many grammatical errors relating to english use and I would suggest that the manuscript would benefit from proof reading by an editor with english as a first language, this is minor and the manuscript was comprehensible.
Response; We have used an editing service for extensive English revisions.
Reviewer 2 Report
The study investigates the significance and characteristics of REM reltaed OSA, using an appropriate defintition and sound selection criteria.
My only concern is related with the English writing and clarity.
Below are a few examples of sentences that need to be improved.
line 60 materials methods: all subjects were of Korean....what?
line 102 (results): Female was more prevalent in REM-OSA patients" Female sex?
line 190: "The difference between both groups in relation to AI was even bigger as eightfold "
line 195: In present study, most compositions of sleep architecture were not significantly different except stage N1 sleep between both groups.
line 250
lease clarify this sentence
In conclusion, although the data which included in this study have longer duration 250 of TST and duration of REM sleep than previous studies, the prevalence of REM-OSA 251 similar with previous finding in Korean study and we found that REM-OSA is frequent 252 in mild to moderate range of OSA and females in Korean population as shown before in 253 Western studies.
Author Response
The study investigates the significance and characteristics of REM reltaed OSA, using an appropriate defintition and sound selection criteria.
My only concern is related with the English writing and clarity.
Response; We have used an editing service for extensive English revisions as below you commented.
Below are a few examples of sentences that need to be improved.
line 60 materials methods: all subjects were of Korean....what?
Response: All subjects were of Korean. → All subjects were Korean.
line 102 (results): Female was more prevalent in REM-OSA patients" Female sex?
Response: 53.6% of REM-OSA patients were females, while in the nREM-OSA group there was a lower prevalence of females (21.7%). Female was more prevalent in REM-OSA patients (53.6% vs. 21.7%, p < 0.001). → 53.6% of the REM-OSA patients were female, whereas the nREM-OSA group had a lower prevalence of females (21.7%, p < 0.001).
line 190: "The difference between both groups in relation to AI was even bigger as eightfold "
Response; . The difference between both groups in relation to AI was even bigger as eightfold. → The difference between the groups in AI was as high as eightfold.
line 195: In present study, most compositions of sleep architecture were not significantly different except stage N1 sleep between both groups.
Response; In present study, most compositions of sleep architecture were not significantly different except stage N1 sleep between both groups. → In this study, most components of sleep architecture (except stage N1 sleep) showed no significant differences between the groups.
line 250 Please clarify this sentence
In conclusion, although the data which included in this study have longer duration of TST and duration of REM sleep than previous studies, the prevalence of REM-OSA similar with previous finding in Korean study and we found that REM-OSA is frequent in mild to moderate range of OSA and females in Korean population as shown before in Western studies.
Response; In conclusion, although the data which included in this study have longer duration of TST and duration of REM sleep than previous studies, the prevalence of REM-OSA similar with previous finding in Korean study and we found that REM-OSA is frequent in mild to moderate range of OSA and females in Korean population as shown before in Western studies. → In conclusion, although the data which included in this study have longer duration of TST and duration of REM sleep than previous studies, the prevalence of REM-OSA similar with previous finding in Korean study. And, as shown before in Western studies, we also found that REM-OSA group had more prevalence of females and REM-OSA is more commonly diagnosed in patients with mild to moderate OSA than nREM-OSA in Korean population.
Round 2
Reviewer 1 Report
For the reasons the authors have stated, they cannot improve upon design which unfortunately makes my original comments still relevant.
They are not able to give much medical information about the cohort, they have not followed up the group or looked at whether they were issued with CPAP. There simply isn't a good reason given and the referral pathway not stated. Not checking supine AHI is simply unusual and most standard software can report this, and therefore with respect they have chosen not to go back and analyse but it would remain of benefit.
This remains a large sample of convenience where there is much missing data and I do not feel there are novel findings, fundamentally this is a flawed design.
There are again multiple minor errors that would require review by a native English speaker, this remains minor but present.
Author Response
I am sorry that I sent you manuscript without marking English revision for major revision. It was my mistake. This time I submit marked version of manuscript with extensive English revision that I should have submitted already.
For the reasons the authors have stated, they cannot improve upon design which unfortunately makes my original comments still relevant.
They are not able to give much medical information about the cohort, they have not followed up the group or looked at whether they were issued with CPAP. There simply isn't a good reason given and the referral pathway not stated.
Response; As I described before, this study is retrospective. Tertiary clinics conducted PSG primarily in Korea, and there was no systematic medical process to manage patients in long-term. Thus, despite our clinics were tertiary, referral information and outcomes of patients treated with CPAP were not available. I am asking your understanding for this limitation.
Not checking supine AHI is simply unusual and most standard software can report this, and therefore with respect they have chosen not to go back and analyse but it would remain of benefit.
Response; I checked supine AHI and analyzed it. The REM-OSA group had less supine AHI than the mild and moderate nREM-OSA group. I revised tables in the revised version of the manuscript and described it in the results of the manuscript.
This remains a large sample of convenience where there is much missing data and I do not feel there are novel findings, fundamentally this is a flawed design.
Response: Actually, I acknowledge that there was no remarkable difference between this study and previous studies. In this study, we just used data with longer TST and REM sleep time compared to previous studies.
There are again multiple minor errors that would require review by a native English speaker, this remains minor but present.
Response: This revised version of manuscript is reviewed by native English speaker, and it is marked on the revised version of manuscript.